# Noradrenaline Synergistically Enhances *Porphyromonas gingivalis* LPS and OMV-Induced Interleukin-1*β* Production in BV-2 Microglia Through Differential Mechanisms

**DOI:** 10.3390/ijms26062660

**Published:** 2025-03-15

**Authors:** Sakura Muramoto, Sachi Shimizu, Sumika Shirakawa, Honoka Ikeda, Sayaka Miyamoto, Misato Jo, Uzuki Takemori, Chiharu Morimoto, Zhou Wu, Hidetoshi Tozaki-Saitoh, Kosuke Oda, Erika Inoue, Saori Nonaka, Hiroshi Nakanishi

**Affiliations:** 1School of Pharmacy, Yasuda Women’s University, Hiroshima 731-0153, Japan; 22141152@st.yasuda-u.ac.jp (S.M.); 20141218@st.yasuda-u.ac.jp (S.S.); 22141124@st.yasuda-u.ac.jp; (S.S.); 22141203@st.yasuda-u.ac.jp (H.I.); 20141243@st.yasuda-u.ac.jp (S.M.); 20141219@st.yasuda-u.ac.jp (M.J.); 22141232@st.yasuda-u.ac.jp (U.T.); 22141153@st.yasuda-u.ac.jp (C.M.); 19141105@st.yasuda-u.ac.jp (E.I.); 2Department of Aging Science and Pharmacology, Faculty of Dental Science, Kyushu University, Fukuoka 812-8582, Japan; zhouw@dent.kyushu-u.ac.jp; 3Department of Pharmaceutical Sciences, School of Pharmacy at Fukuoka, International University of Health and Welfare, Okawa 831-8501, Japan; saitoh@iuhw.ac.jp; 4Department of Pharmaceutical Chemistry, Faculty of Pharmacy, Yasuda Women’s University, Hiroshima 731-0153, Japan; 5Department of Pharmacology, Faculty of Pharmacy, Yasuda Women’s University, Hiroshima 731-0153, Japan; nonaka-s@yasuda-u.ac.jp

**Keywords:** activator protein 1, BV2 microglia, interleukin-1*β*, lipopolysaccharide, noradrenaline, nuclear factor-*κ*B, outer membrane vesicles, *Porphyromonas gingivalis*, synergistic augmentation

## Abstract

Infection with *Porphyromonas gingivalis* (*Pg*), which is a major periodontal pathogen, causes a large number of systemic diseases based on chronic inflammation such as diabetes and Alzheimer’s disease (AD). However, it is not yet fully understood how *Pg* can augment local systemic immune and inflammatory responses during progression of AD. There is a strong association between depression and elevated levels of inflammation. Noradrenaline (NA) is a key neurotransmitter that modulates microglial activation during stress conditions. In this study, we have thus investigated the regulatory mechanisms of NA on the production of interleukin-1*β* (IL-1*β*) by microglia following stimulation with *Pg* virulence factors, lipopolysaccharide (LPS), and outer membrane vesicles (OMVs). NA (30–1000 nM) significantly enhanced the mRNA level, promoter activity, and protein level of IL-1*β* up to 20-fold in BV-2 microglia following treatment with *Pg* LPS (10 μg/mL) and OMVs (150 μg of protein/mL) in a dose-dependent manner. Pharmacological studies have suggested that NA synergistically augments the responses induced by *Pg* LPS and OMVs through different mechanisms. AP-1 is activated by the *β*_2_ adrenergic receptor (A*β*_2_R)-mediated pathway. NF-*κ*B, which is activated by the *Pg* LPS/toll-like receptor 2-mediated pathway, is required for the synergistic effect of NA on the *Pg* LPS-induced IL-1*β* production by BV-2 microglia. Co-immunoprecipitation combined with Western blotting and the structural models generated by AlphaFold2 suggested that cross-coupling of NF-*κ*B p65 and AP-1 c-Fos transcription factors enhances the binding of NF-*κ*B p65 to the I*κ*B site, resulting in the synergistic augmentation of the IL-1*β* promoter activity. In contrast, OMVs were phagocytosed by BV-2 microglia and then activated the TLR9/p52/RelB-mediated pathway. The A*β*_2_R/Epac-mediated pathway, which promotes phagosome maturation, may be responsible for the synergistic effect of NA on the OMV-induced production of IL-1*β* in BV-2 microglia. Our study provides the first evidence that NA synergistically enhances the production of IL-1*β* in response to *Pg* LPS and OMVs through distinct mechanisms.

## 1. Introduction

*Porphyromonas gingivalis* (*Pg*) is a Gram-negative bacterium and a major pathogen of periodontal disease [1]. Accumulating epidemiological evidence indicates that infection with *Pg*, which is a major periodontal pathogen, causes a large number of systemic diseases, including diabetes mellitus, rheumatoid arthritis, and Alzheimer’s disease (AD) [2]. Recently, it has been reported that approximately 40% of patients diagnosed with AD lack AD pathology including the cerebrospinal fluid levels of amyloid-*β*42 and phosphorylated tau [3]. Therefore, microglia-mediated neuroinflammation is considered a key driver of dementia associated with a variety of diseases including AD. Importantly, *Pg* can influence the development of these periodontitis-related systemic disorders by affecting the adaptive immune responses [2]. In contrast, *Pg* lipopolysaccharide (LPS) is a relatively poor inducer of monocyte production of proinflammatory cytokines compared with *E. coli* LPS [4,5]. *Pg* LPS stimulated an inflammatory response when injected into connective tissue but was minimally stimulatory when a systemic response was measured [6,7]. Furthermore, *Pg* LPS is a weaker toll-like receptor (TLR) 2/4 agonist and NF-κB/STAT signaling activator in BV-2 microglia compared with *E. coli* LPS [8]. The presence of fewer acyl chains and phosphate groups in *Pg* LPS-lipid A than *E. coli*-lipid A may be responsible for a lower level of TLR activating potency of *Pg* LPS. Moreover, *Pg* LPS exerted antagonistic effects toward TLR4-dependent cell activation by *E. coli* LPS [9]. These properties may allow *Pg* to evade innate host defense mechanisms without endotoxin tolerance, leading to chronic inflammation. However, it is not yet fully understood the mechanisms by which *Pg* LPS can augment local systemic immune and inflammatory responses during progression of AD. Besides LPS, *Pg* produces outer membrane vesicles (OMVs) that contain LPS, proteases including gingipains and pathogen-derived DNA and RNA [1]. OMVs secreted from *Pg* induced AD-like pathologies in middle-aged mice [10].

There is a strong association between depression and elevated levels of inflammation. Social stress enhances *Pg* LPS-induced inflammatory responses by CD11b-positive cells [11]. In addition, chronic stress can significantly enhance the pathological progression of periodontitis through an adrenergic signaling-mediated inflammatory response [12]. Moreover, stress-induced microglial activation, anxiety-like behaviors and IL-1*β* production were inhibited by a pharmacological blockade of adrenergic receptors [13,14,15]. We previously reported that noradrenaline (NA) induces the production of IL-1*β* through the activation of *β*_2_ adrenergic receptors (A*β*_2_R)/exchange proteins directly activated by cAMP (Epac), which activates AP-1-mediated transcription [16]. Both NF-κB and AP-1 play important roles as transcription factors in the expression of IL-1*β*, which is a critical step in inflammation through the induction of other proinflammatory cytokines and chemokines [17,18].

Therefore, we hypothesized that NA may enhance neuroinflammation following systemic stimulation with *Pg* virulence factors, because enhanced NA release is a central response to stress. In the present study, we investigated the possible functional interaction between NA and *Pg* virulence factors in the production of IL-1*β* by BV-2 microglia. The present findings highlight that NA synergistically enhances the production of IL-1*β* in response to *Pg* LPS and OMVs through distinct mechanisms. This study is the first to show the existence of an intrinsic augmentation mechanism of *Pg* virulence factor-induced neuroinflammation by microglia, providing biological evidence for the impact of stress on the progression of periodontitis-related systemic inflammatory diseases.

## 2. Results

### 2.1. Synergistic Expression of IL-1β mRNA in BV-2 Microglia by NA and Pg Virulence Factors, Pg LPS and OMVs

We first examined the effects of NA on the *Pg* LPS-induced expression of IL-1*β* by quantitative reverse transcription polymerase chain reaction (qPCR). *Pg* LPS (10 μg/mL) and OMVs (150 μg of protein/mL) induced the expression of IL-1*β* mRNA, as previously reported [19,20] (Figure 1B,C). BV-2 microglia were treated with a combination of NA and *Pg* virulence factors. The treatment of BV-2 microglia with a combination of NA (30–1000 nM) and *Pg* LPS resulted in a significant enhancement of the IL-1*β* mRNA expression by up to 9.0-fold relative to treatment with NA alone or *Pg* LPS alone in a concentration-dependent manner (Figure 1A,B). A combination of both NA and OMVs also led to an increase in the IL-1*β* mRNA expression of up to almost 12.8-fold relative to treatment with NA alone or OMVs alone in a concentration-dependent manner (Figure 1A,C). These observations suggest that NA exhibits a synergistic effect on the *Pg* virulence factor-induced IL-1*β* expression of mRNA expression in BV-2 microglia.

### 2.2. Synergistic Production of IL-1β in BV-2 Microglia Following Treatment with NA and Pg Virulence Factors, Pg LPS, and OMVs

The synergistic effect of NA on the *Pg* virulence factor-induced IL-1*β* mRNA expression in BV-2 microglia may be caused by increased transcription or increased stability of IL-1*β* mRNA in BV-2 microglia by activation of transcription factors and/or kinases. To dissect the mechanism by which NA synergizes with *Pg* virulence factors to upregulate the expression of IL-1*β*, the luciferase activities of NanoLuc (Nluc) reporter BV-2 microglia were measured after treatment with either NA alone or in combination with *Pg* virulence factors. NA (30–1000 nM) induced luciferase activity in a concentration-dependent manner, ranging from 4 to 20 × 10^3^ relative light units (RLU) (Figure 1D). On the other hand, *Pg* LPS (10 μg/mL) and OMVs (150 μg of protein/mL) induced luciferase activities of 10–13 × 10^3^ and 6–13 × 10^3^ RLU, respectively. Treatment of Nluc reporter BV-2 microglia with a combination of NA (30–1000 nM) and *Pg* LPS resulted in the enhancement of *Pg* LPS-induced luciferase activities in a concentration-dependent manner, by up to 22.3-fold relative to *Pg* LPS alone (Figure 1E). On the other hand, NA at concentrations of 300 and 1000 nM significantly increased OMV-induced luciferase activities by 8.1- and 29.7-fold relative to OMVs alone, respectively (Figure 1F).

Next, we performed time-course studies of the synergistic production of IL-1*β* in BV-2 microglia following treatment with NA and *Pg* virulence factors. *Pg* LPS-induced luciferase activity peaked 1 h after application and returned to its basal level within 6 h (Figure 2A). The combined application of NA and *Pg* LPS also peaked at 1 h and returned to its basal level within 6 h (Figure 2B). OMVs alone or in combination with NA and *Pg* LPS also induced luciferase activity, which peaked at 30 min. After 1 h, the mean luciferase activity remained elevated to a similar level and then returned to the basal level within 6 h (Figure 2C,D). These results suggest that the synergistic production of IL-1*β* in BV-2 microglia following treatment with NA and *Pg* virulence factors is manifested at both transcriptional and translational levels.

### 2.3. Production of IL-1β by BV-2 Microglia Following Treatment with NA and Pg Virulence Factors, Pg LPS, and OMVs

The mean levels of IL-1*β* produced by BV-2 microglia were measured following treatment with NA, *Pg* virulence factors, or their combination. We first determined the mean concentrations of IL-1*β* in the cell lysates of BV-2 microglia 1 h after treatment with NA, *Pg* LPS, OMVs, NA plus *Pg* LPS, and NA plus OMVs by immunoassay. Treatment of BV-2 microglia with a combination of both NA and *Pg* LPS resulted in a significant enhancement of IL-1*β* up to almost 10-fold relative to treatment with NA or *Pg* LPS alone (Figure 3A). A combination of both NA and OMVs also led to the enhancement of IL-1*β* (almost 20-fold) relative to treatment with NA alone or OMVs alone (Figure 3B).

IL-1*β* was produced as a proform with an apparent molecular mass of 33 kDa and proIL-1*β* increased after treatment with NA (3 μM), *Pg* LPS (30 μg/mL), or their combined application in BV-2 microglia (Figure 3C). At 1 h after treatment, the 17 kDa mature form of IL-1*β* was visible in the cell lysates (Figure 3C). The amounts of both the 33 kDa precursor (proIL-1*β*) and 17 kDa mature IL-*β* were markedly increased in the cell lysates of BV-2 microglia after stimulation with NA plus *Pg* LPS relative to stimulation with either NA alone or *Pg* LPS alone (Figure 3C). Interestingly, the amounts of proIL-1*β* and an unconventional 20 kDa form of IL-1*β* were markedly increased in the cell lysates of BV-2 microglia after stimulation with NA plus OMVs compared to stimulation with either NA or OMVs alone (Figure 3D).

### 2.4. Effects of Inhibitors of NF-κB and AP-1 Signaling Pathways on the Production of IL-1β After Stimulation with NA + Pg LPS or NA + OMVs

NF-*κ*B and AP-1 are key transcription factors that orchestrate the expression of several genes involved in inflammation. Although regulated by different mechanisms, they appear to be activated simultaneously by a multitude of stimuli. To address possible roles of these signaling molecules in synergistic effect of NA and *Pg* virulence factors on the IL-1*β* production in BV-2 microglia, we examined the effects of various inhibitors of molecules that are involved in NF-*κ*B and AP-1 signaling pathways.

Wedelolactone (30 μM), an I*κ*B kinase (IKK) inhibitor, almost completely abolished the luciferase activity induced by NA (1 μM), *Pg* LPS (10 μg/mL), or their combined application (Figure 4A). An NF-*κ*B activation inhibitor (NAI, 100 nM) also significantly suppressed NA plus *Pg* LPS-induced luciferase activities by approximately 80% (Figure 4B). The I*κ*B kinase inhibitor wedelolactone also almost completely abolished luciferase activity induced by either NA or NA plus OMVs (150 μg of protein/mL), while wedelolactone failed to inhibit OMV-induced luciferase activity (Figure 4C). In contrast, NAI failed to inhibit the luciferase activity induced by either OMVs or NA plus OMVs (Figure 4D).

The MEK1/2 inhibitor U0126 (15 μM) significantly suppressed NA plus *Pg* LPS-induced luciferase activity (Figure 5A). The AP-1 inhibitor SR11302 (10 μM) significantly suppressed NA + *Pg* LPS-induced luciferase activity by approximately 20% (Figure 5B). U0126 (15 μM) and SR11302 (10 μM) failed to suppress OMV- or NA plus OMV-induced luciferase activity (Figure 5C,D).

### 2.5. Effects of Inhibitors of Aβ_2_R/Epac- and TLR29/-p52/RelB-Mediated Pathways on the Production of IL-1β After Stimulation with NA or Pg LPS

NA induces the production of IL-1*β* by BV-2 microglia through an A*β*_2_R/Epac-mediated pathway [16]. Epac activates AP-1-mediated transcription. We examined the effects of a selective A*β*_2_R inhibitor, ICI-118551, on the synergistic effect of NA and *Pg* virulence factors on the production of IL-1*β* in BV-2 microglia. ICI-118551 (0.1 μM) significantly suppressed the production of IL-1*β* in BV-2 microglia following stimulation with NA (1 μM) plus *Pg* LPS (10 μg/mL) or NA (1 μM) plus OMVs (150 μg of protein/mL) (Figure 6A,B).

When NA at concentration of 1 μM was used, ICI-118551 significantly inhibited the production of IL-1*β* in BV-2 microglia stimulated with NA plus *Pg* virulence factors. However, ICI-118551 failed to inhibit the NA-induced production of IL-1*β* (Figure 6A,B). The mean value decreased; however, the difference was not statistically significant. Furthermore, NAI significantly inhibited the production of IL-1*β* in BV-2 microglia stimulated with NA plus *Pg* LPS, but not with *Pg* LPS alone (Figure 4B). Similar results were obtained using U0126 and SR11302 (Figure 5A,B).

This discrepancy may be due to the different mean luciferase activities. NA (1 μM) alone or *Pg* LPS (10 μg/mL) alone induced luciferase activity of up to 20 × 10^3^ RLU, whereas the combined application induced luciferase activity of up to 300 × 10^3^ RLU. Therefore, we stimulated BV-2 microglia with NA (3 μM) or *Pg* LPS (30 μg/mL). ICI-118551 significantly suppressed the production of IL-1*β* by BV-2 microglia following stimulation with NA (3 μM), but not by *Pg* LPS (30 μg/mL) (Figure 6C,D). In contrast, NAI, U0126, and SR11302 significantly suppressed the NA- or *Pg* LPS-induced production of IL-1*β* (Figure 6C,D).

The possible involvement of phagocytosis-related pathways in the production of IL-1*β* by BV-2 microglia following OMV treatment was further examined. The mean values of the relative luciferase activities of the IL-1*β* probe induced by NA, OMVs, and their combined application were significantly suppressed by either the TLR9 inhibitor E6446 or the p52 inhibitor SN52 (Figure 6E,F).

### 2.6. Possible Changes in Expression Levels and Protein Complex Formation of NF-κB p65 and AP-1 c-Fos in BV-2 Microglia Following Combined Treatment with NA and Pg LPS

We addressed whether the synergistic effect of NA and *Pg* virulence factors increased the expression levels of NF-*κ*B and AP-1 in addition to their cooperation in the IL-1*β* promoter. The protein level of c-Fos in BV-2 microglia was markedly increased following treatment with NA and *Pg* LPS, whereas the protein level of p65 was unchanged (Figure 7A). Thus, the increased expression of c-Fos may be responsible for the synergistic effects of NA and *Pg* LPS on the production of IL-1*β* in BV-2 microglia through increased AP-1 promoter activity. However, our pharmacological studies showed that the NA plus *Pg* LPS-induced production of IL-1*β* was more effectively suppressed by an NF-*κ*B inhibitor (approximately 80%) than an AP-1 inhibitor (approximately 20%). Therefore, it is likely that increased c-Fos may physically bind to p65 rather than the AP-1 promoter site, because c-Jun and c-Fos are capable of physically interacting with p65, exhibiting potentiated biological activity [21,22,23,24,25,26]. Next, we conducted co-immunoprecipitation combined with Western blotting using anti-p65 and anti-c-Fos antibodies. Co-immunoprecipitation combined with Western blotting showed that c-Fos was present in the protein complexes immunoprecipitated using anti-p65 antibodies in BV-2 microglia stimulated with *Pg* LPS and NA (Figure 7B,C). In Figure 7B, a band corresponding to p65 was clearly visible in the lane for anti-p65 IgG, but not in the lane for normal IgG. In Figure 7C, however, rather multiple smeared bands at 50–60 kDa were visible in the lane for anti-p65 antibody, but not in the lane for normal IgG, probably because a rabbit polyclonal antibody for c-Fos (ab190289, Abcam), which detects multiple bands at 50–60 kDa corresponding to different isoforms of c-Fos, was used in this study.

### 2.7. Prediction of c-Fos Binding to p65

To examine whether c-Fos directly binds to p65, a high-quality model of p65:p50 heterodimer bound to c-Fos was generated using AlphaFold 2 (Figure 8A). In the model, c-Fos forms a complex with p65:p50 heterodimer via the interaction between the leucine-zipper domain of c-Fos and the dimerization domain of p65 with a high reliability (Figure 8A,B). When the structure of p65:p50 heterodimer bound to c-Fos was superimposed on that of p65:p50 heterodimer bound DNA, the leucine-zipper domain of c-Fos could be well inserted in a major groove of DNA (Figure 8C). Therefore, a co-operative complex of NF-*κ*B and AP-1 on DNA stabilizes the p65:DNA complex.

## 3. Discussion

In this study, NA augmented *Pg* virulence factor-induced IL-1*β* transcriptional activities up to 20-fold without affecting either peak or duration time. Considering that higher amounts of NA are released from nonsynaptic varicosities’ noradrenergic terminals, especially under stress conditions, A*β*_2_R on microglia surrounding noradrenergic terminals can be activated. To the best of our knowledge, this is the first report showing that NA synergizes with *Pg* virulence factors to upregulate the expression of IL-1*β* in microglia. The present findings may provide biological evidence for the impact of stress on the progression of periodontitis-related systemic inflammatory diseases.

We recently reported that the delivery of gingipains into cerebral microvascular endothelial cells, probably through OMVs, may be responsible for blood–brain barrier damage through intracellular degradation of tight junction proteins ZO-1 and occludin [28]. Therefore, circulating *Pg* virulence factors may be augmented by NA in the brain parenchyma. Several studies have reported anti-inflammatory effects of NA, which is contrary to our findings [29,30]. The anti- and pro-inflammatory effects of NA are intriguing paradoxes. One suggested common aspect to the abovementioned studies is that experiments showing an anti-inflammatory effect of NA are mostly conducted in the combination with *E. coli* LPS treatment, which induces the translocation of NF-*κ*B into the nucleus [31]. However, we observed that NA augmented the *Pg* LPS-induced production of IL-1*β*. We previously reported that *Pg* LPS-induced luciferase activity in BV-2 microglia was almost completely suppressed by a specific TLR2 antagonist, C29, but not by a specific TLR4 antagonist, TAK-242 [20]. It has been reported that the signaling via TLR2 could be due to PG1828, a lipoprotein synthesized by *Pg* [32,33]. These observations indicate that *Pg* LPS-induced luciferase activity in BV-2 microglia is mediated by lipoproteins via TLR2. Therefore, this discrepancy may be due to the difference in TLRs activated by *Pg* lipoproteins and *E. coli* LPS. Further studies are needed to fully understand the paradoxical effects of NA.

NF-*κ*B and AP-1 play important roles as transcription factors in the expression of various genes induced by bacterial LPS, including *Pg* LPS. Therefore, using specific inhibitors, we investigated the involvement of these transcription factors in the production of IL-1*β* following treatment with NA, *Pg* LPS, and their combination. NAI and SR11302 inhibited the mean levels of luciferase activity induced by NA alone or *Pg* LPS alone by approximately 30–40%. In contrast, NAI and SR11302 inhibited the mean levels of luciferase activity induced by NA plus *Pg* LPS by approximately 80% and 20%, respectively. Therefore, the increased promoter activity of the *κ*B sites is considered to be responsible for the synergistic effect of NA and *Pg* LPS. A typical structure of NF-*κ*B is the p50-p65 dimer (NF-*κ*B1/RelA), because dimer formation is necessary for DNA binding. The *N*-terminal regions of the dimer are responsible for specific DNA contact. In contrast, the *C*-terminal regions are usually highly conserved, and are responsible for dimerization and nonspecific DNA phosphate contact. AP-1 transcription factors are dimers composed of members of the c-Fos and c-Jun protein families. AP-1 c-Fos can enhance the NF-*κ*B target gene expression by binding to AP-1 sites. Moreover, the basic zipper domains of AP-1 c-Jun and c-Fos physically interact with the Rel-homology domain of p65, forming a transcription complex with enhanced DNA binding and potentiated biological functions [21]. Considering that the inhibitory effect of SR11302, which blocks the binding of AP-1 to the promoter region, was lower than NAI, AP-1 may physically interact with p65 to enhance its binding of p65 to the *κ*B sites. We found that c-Fos directly binds to p65 co-immunoprecipitation analyses following co-treatment with NA and *Pg* LPS. Furthermore, the structural models suggest that the binding of c-Fos to the p65:p50 heterodimer induces a conformation change of DNA to form hydrogen bonds between c-Fos and DNA, which is likely to enhance the stabilization of the protein–DNA complex. Persistent NF-*κ*B activation is a hallmark of chronic inflammatory diseases, including neurodegenerative diseases, cancer, and arthritis [34,35].

On the other hand, OMVs can also elicit various inflammatory and immune responses [36]. *Pg* LPS and gingipains are directly detected by TLR2 and protease-activated receptor 2 on the cell membrane, respectively [37]. In the present study, NAI and SR11302 showed no or moderate inhibitory effects on luciferase activity induced by OMVs alone and OMVs plus NA, suggesting that OMVs and OMVs plus NA activate transcriptional pathways other than either NF-*κ*B p65 or AP-1. OMVs are quickly phagocytosed by BV-2 microglia, and the phagocytic/endocytic pathway is required for the production of IL-1*β* [20]. It was also noted that the amounts of proIL-1*β* and an unconventional 20 kDa form of IL-1*β* were markedly increased in the cell lysates of BV-2 microglia after stimulation with NA plus OMVs relative to stimulation with either NA or OMVs alone. LPS treatment produces the 20 kDa form of IL-1*β* intracellularly [38]. In contrast, leaked proIL-1*β* from LPS-primed microglia after ATP stimulation is processed into the 20 kDa form of IL-1*β* by cathepsin D [39,40]. Interestingly, the formation of the 20 kDa form of IL-1*β* prevents the generation of mature IL-1*β* and thus may limit inflammation, because the 20 kDa form of IL-1*β* is minimally active at IL-1R1 and is not further cleaved to highly active 17 kDa mature form [40].

RNA and DNA in bacterial OMVs are recognized by TLR7/TLR8 and TLR9 in phago-endosomes, respectively. We found that the TLR9 inhibitor E6446 significantly suppressed the luciferase activity induced by both OMVs alone and NA plus OMVs. Moreover, the p52 inhibitor SN52 significantly suppressed the luciferase activity induced by both OMVs alone and NA plus OMVs. OMVs are thought to induce inflammatory and immune responses through the activation of the TLR9-p52/RelB-mediated pathway after phagocytosis. However, the mechanisms for the NA plus OMV-induced synergistic effect remain unclear in this study. It is likely that the A*β*_2_R/Epac-mediated pathway enhances trafficking of endosomes and/or phagosomes to promote their fusion, because Epac-Rap1 signaling was suggested to play an important role in the interaction between late endosomes and phagosomes [41].

Additional experiments are necessary to further elucidate physical and functional interactions between NF-*κ*B and AP-1 c-Fos that lead to enhanced transcriptional activity at mainly the *κ*B enhancer-dependent promoter. Moreover, the mechanisms by which these two transcriptional factors are differentially regulated by NA and *Pg* virulence factors are to be addressed in future studies.

The present study is beset with certain limitations. First, all experimental findings were obtained from in vitro studies by use of BV-2 microglia. Therefore, additional experiments using animals subjected to stress are needed to verify the present findings. Second, we used only commercially available *Pg* LPS (STD PG-LPS, cat: tlrl-pglps, InvitroGen), which was obtained by classical methods using hot-phenol extraction using the bacterial strain ATCC 33277, suggesting that the TLR2-dependent IL-1*β* production by BV-2 microglia was mediated mainly by concomitant lipoproteins. Furthermore, *Pg* LPS-lipid A may be received modification of acetylchains and phosphates of lipid A during preparation of *Pg* LPS, resulting in reduction of TLR4 activating potency. Different bacterial strains contain different factors in their own OMVs, which may determine their pathogenicities [42]. The ATCC 33277 used in this study is classified as a less-virulent strain. Further studies are also necessary to examine the effects of *Pg* LPS, lipoproteins, and OMVs prepared from several more virulent bacterial strains, including W83 and possibly more virulent clinical isolates, on TLRs of microglia.

## 4. Materials and Methods

### 4.1. Reagents

NA, U0126, a MEK inhibitor, and NAI, an inhibitor of NF-*κ*B p65 transcriptionally activity, were purchased from Sigma-Aldrich (St. Louis, MO, USA). Wedelolactone, an IKK inhibitor, was purchased from Tokyo Chemical Industry Co., Ltd. (Tokyo, Japan). SR11302, an inhibitor of AP-1 c-Fos transcriptional activity, was purchased from MedChem Express (Monmouth Junction, NJ, USA). *Pg* LPS (PG-LPS, cat: tlrl-pglps) was purchased from InvivoGen (San Diego, CA, USA).

### 4.2. Cell Culture

The BV-2 cells, a murine microglial cell line [43], and a well-accepted alternative to primary microglia [44,45], were used in this study. BV-2 microglia were cultured in Dulbecco’s modified Eagle’s medium (Thermo Fisher Scientific, Waltham, MA, USA) supplemented with 5% fetal bovine serum (Nichirei, Tokyo, Japan), penicillin-streptomycin (Fujifilm, Tokyo, Japan). BV-2 cells at passage number 3–10 were used in the experiments. To establish Nluc probe-expressing cells (Nluc reporter BV-2 microglia), we infected the cells with a lentiviral vector carrying the Nluc probe, as described previously [16]. The Nluc luciferase protein was retained only when the proteolytic processing of IL-1*β* was successful, as it was able to escape proteasome degradation.

### 4.3. Measurement of the Luciferase Activity (RLU)

Nluc reporter BV-2 microglia were plated in 96-well culture plates (IWAKI, Tokyo, Japan) at a density of 5 × 104 cells per well. After overnight culture, drug treatments were performed, and luciferase activity following treatment with NA (1 or 3 µM), *Pg* LPS (10 or 30 µg/mL) or OMVs (150 µg of protein/mL) was measured using a luminometer (GloMax; Promega Corp., Madison, WS, USA) with a Nano-Glo^®^ luciferase assay system (N1110; Promega Corp.) according to the manufacturer’s instructions. RLU induced by NA*, Pg* LPS, or OMVs was then measured in sum of cells and culture media.

### 4.4. Bacterial Culture and Isolation of OMVs

Pg ATCC 33277 was maintained as previously described [28]. Detailed methods for preparing OMVs were previously reported [28]. Briefly, culture mediums of Pg were centrifuged to remove bacterial cells. The supernatant was filtrated with Millipore filter (pore size 0.22 μm, Millipore, MA, USA), and then ultracentrifuged. The pellet was collected as OMVs and dissolved in phosphate-buffered saline. The protein concentration of OMVs were measured by Pierce™ BCA protein assay kit (Thermo Fisher Scientific Inc., Waltham, MA, USA) and the concentrations of OMVs were adjusted to 150 μg of protein/mL.

### 4.5. RT-qPCR

The mRNA levels of IL-1*β* expressed in BV-2 microglia were measured by RT-qPCR, as previously described [19,46]. The primer pair sequences were as follows: IL-1*β*, 5′-CAACCAACAAGTGATATTCTCCAT-3′ and 5′-GAT CCA CACTCTCCAGCTGCA-3′; *β*-actin, 5′-GGCATTGTGATGGACTCCG-3′ and 5′-GCTGGAAGGTGGACAGTGA-3′. For data normalization, an internal control (*β*-actin) was used for cDNA input, and the relative units were calculated using a calibration curve method.

### 4.6. Immunoassay for IL-1β

The concentrations of IL-1*β* in the cell lysates of BV-2 microglia were measured using a LumitTM IL-1*β* (Mouse) immunoassay (Promega Corp., Madison, WS, USA) 1 h after treatment with NA, *Pg* LPS, OMVs, NA plus *Pg* LPS, and NAplus OMVs, according to the manufacturer’s instructions. Luminescence generated, which was proportional to the amount of IL-1*β*, was measured using a multimode plate reader EnSight (Parkin-Elmer, Shelton, CT, USA).

### 4.7. Immunoblotting

BV-2 microglia were seeded in a 6 cm Petri dish (Sumitomo Bakelite, Tokyo, Japan) at a density of 3.3 × 106 cells/dish for 1 day. After treatment with NA (3 µM), *Pg* LPS (30 µg/mL) or OMVs (150 µg protein/mL), and their combination, each specimen was electrophoresed using 15% SDS-polyacryl-amide gels, and then proteins on the SDS gels were electrophoretically transferred to nitrocellulose membranes, and immunoblotted as previously described [17,46]. The primary antibodies used were as followings: goat anti-IL-1*β* antibodies (1:1000; R&D systems, Minneapolis, MN, USA); rabbit anti-glyceraldehyde-3-phosphate dehydrogenase (GAPDH) antibodies (1:1000; Proteintech, Tokyo, Japan).

### 4.8. Co-Immunoprecipitation and Immunoblot Analysis

BV-2 microglia were seeded in a 10 cm Petri dish (IWAKI, Tokyo, Japan) at a density of 3.3 × 106 cells/dish overnight. After treatment with NA (3 µM), *Pg* LPS (30 µg/mL), and their combination, cells were collected and cell lysates (1 mg of protein/each) were performed immuno-precipitation using a Pierce co-immunoprecipitation kit (Thermo Fischer Scientific) according to manufacture’s instructions. The antibodies for immunoprecipitation were rabbit anti-NF-*κ*B p65 antibodies (20 µL; ab2071297, Abcam, Tokyo, Japan). Immuno-precipitates were separated on a 10% SDS-PAGE gel, transferred to PVDF membrane (Merck, Boston, MA, USA), and immunoblotted. The primary antibodies for immunoblotting used were as follows: rabbit anti-c-Fos antibodies (1:1000; ab190289, Abcam); rabbit anti-NF-*κ*B p65 anti-bodies (1:1000; Ab16502, Abcam).

### 4.9. AlphaFold Predictions

The AlphaFold models of the ternary complex consisting of the DNA binding domain and dimerization domain of p65 (aa 1–291), those of p50 (aa 1–352), and full length of c-Fos were predicted using the AlphaFold v2.0 algorithm on the Co-lab server (https://colab.research.google.com/github/sokrypton/ColabFold/blob/main/AlphaFold2.ipynb, accessed on 20 November 2023) [47]. Predictions were performed with default multiple sequence alignment generation using the MMSeqs2 server, with 48 recycles and templates (homologous structures). The best of the five predicted models (rank 1) computed by AlphaFold was considered in the present work. The structure of the complex consisting of p65, p50, c-Fos, and DNA was generated based on the AlphaFold model of p65:p50 hetero-dimer bound to leucine-zipper domain of c-Fos and the structure of DNA-bound p65:p50 heterodimer (PDB code 2I9T).

### 4.10. Statistical Analyses

Data are presented as the mean ± SE. The results were analyzed by a one-way analysis of variance (ANOVA) with a post hoc Tukey’s test using the GraphPad Prism8 (GraphPad Software, Inc., CA, USA) software package. *p* values of <0.05 were considered to indicate statistical significance.

## 5. Conclusions

It was concluded that stress-induced upregulation of NA and Pg virulence factors can synergistically augment the production of IL-1β by microglia. The present study is the first to show that stress-induced upregulation of NA and Pg virulence factors can synergistically augment the production of IL-1β by microglia. This may explain the reason why stress influences the progression of periodontitis-related systemic inflammatory diseases.

## Figures and Tables

**Figure 1 ijms-26-02660-f001:**
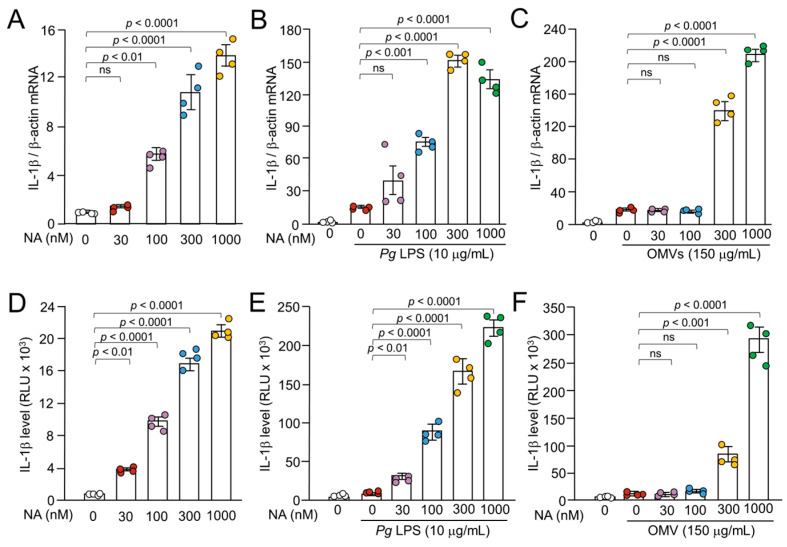
Synergistic effects of NA on the *Pg* virulence factor induced the expression of IL-1*β* by BV-2 microglia. (**A**–**C**) The mean relative level of IL-1*β* mRNA following treatment with NA, NA + *Pg* LPS, and NA + OMVs for 1 h. (**A**) The mean relative level of IL-1*β* mRNA induced by NA with various concentrations (30–1000 nM). (**B**,**C**) The mean relative level of IL-1*β* mRNA induced by *Pg* LPS (10 μg/mL) (**B**) or OMVs (150 μg of protein/mL) (**C**) in the absence or presence of NA (30–1000 nM). (**D**–**F**) The mean relative luciferase activity of the IL-1*β* probe following treatment with NA, NA + *Pg* LPS, and NA + OMVs for 1 h. (**D**) The mean relative luciferase activity of the IL-1*β* probe induced by NA with various concentrations (30–1000 nM). (**E**,**F**) The mean relative luciferase activity of the IL-1*β* probe induced by *Pg* LPS (10 μg/mL) (**E**) or OMVs (150 μg of protein/mL) (**F**) in the absence or presence of NA (30–1000 nM). Data relative to the values are presented as the mean ± standard error (SE) of 4 independent experiments. ns: not statistically significant.

**Figure 2 ijms-26-02660-f002:**
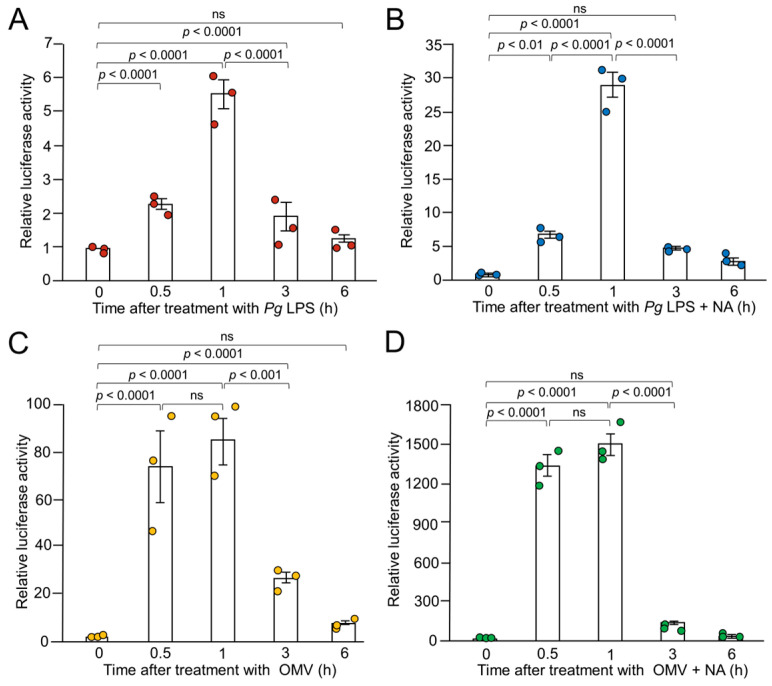
Time course of the mean relative luciferase activity of the IL-1*β* probe induced by *Pg* LPS, OMVs, and their combination with NA. (**A**,**B**) The mean relative luciferase activity of the IL-1*β* probe induced by either *Pg* LPS alone (10 μg/mL) (**A**) or combination with NA (1 μM) (**B**). (**C**,**D**) The mean relative luciferase activity of the IL-1*β* probe induced by either OMVs alone (150 μg of protein/mL) (**C**) or combination with NA (1 μM) (**D**). Data relative to the values are presented as the mean ± SE of 3 independent experiments. ns: not statistically significant.

**Figure 3 ijms-26-02660-f003:**
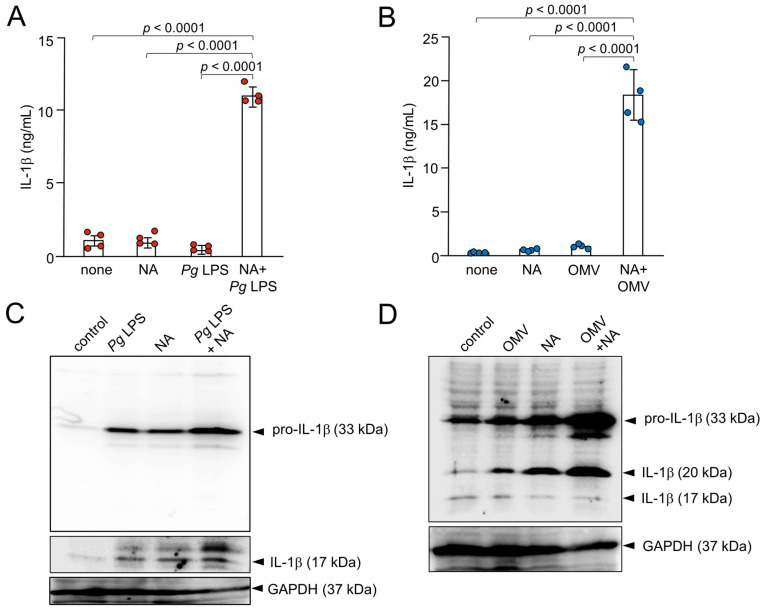
Production of IL-1*β* by BV-2 microglia following treatment with NA, *Pg* virulence factors, and their combination. (**A**) The mean concentration IL-1*β* in the cell lysates of BV-2 microglia following treatment with NA (1 μM), *Pg* LPS (10 μg/mL), and their combined application. (**B**) The mean concentration of IL-1*β* in the cell lysates of BV-2 microglia following treatment with NA (1 μM), OMVs (150 μg of protein/mL), and their combined application. Data are presented as the mean ± SE of 4 independent experiments. (**C**) Cell lysates from BV-2 microglia following treatment with NA (3 μM), *Pg* LPS (30 μg/mL), or their combined application for 1 h were subjected to immunoblotting of IL-1*β*. (**D**) Cell lysates from BV-2 microglia following treatment with NA (3 μM), OMVs (150 μg of protein/mL), or their combined application for 1 h were subjected to immunoblotting of IL-1*β*. Immunoblotting of GAPDH was used as a loading control.

**Figure 4 ijms-26-02660-f004:**
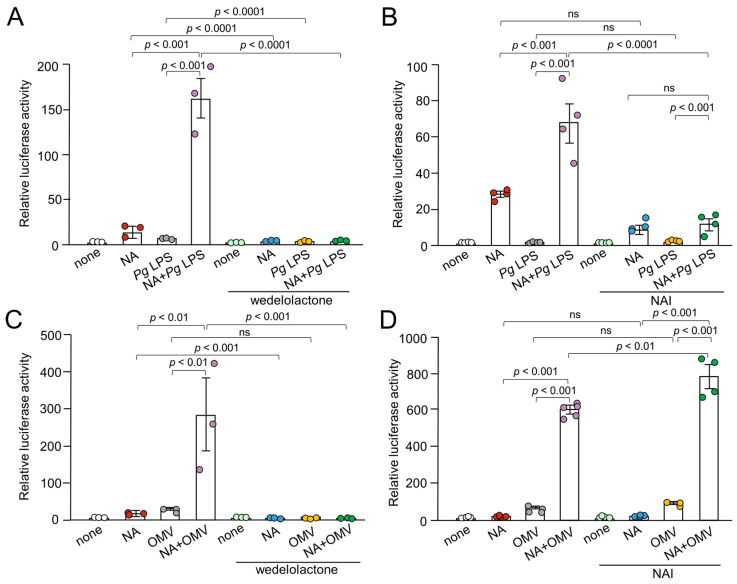
Effects of NF-*κ*B signaling pathway inhibitors on the production of IL-1*β* by BV-2 microglia following treatment with NA, *Pg* virulence factors, and their combination. (**A**,**B**) The mean relative luciferase activity of the IL-1*β* probe induced by NA (1 μM), *Pg* LPS (10 μg/mL), and their combined application in the absence or presence of wedelolactone (30 μM) (**A**) or NAI (100 nM) (**B**). (**C**,**D**) The mean relative luciferase activity of the IL-1*β* probe induced by NA (1 μM), OMVs (150 μg/mL) and their combined application in the absence or presence of wedelolactone (30 μM) (**C**) or NAI (100 nM) (**D**). Data relative to the values are presented as the mean ± SE of 3–5 independent experiments. ns: not statistically significant.

**Figure 5 ijms-26-02660-f005:**
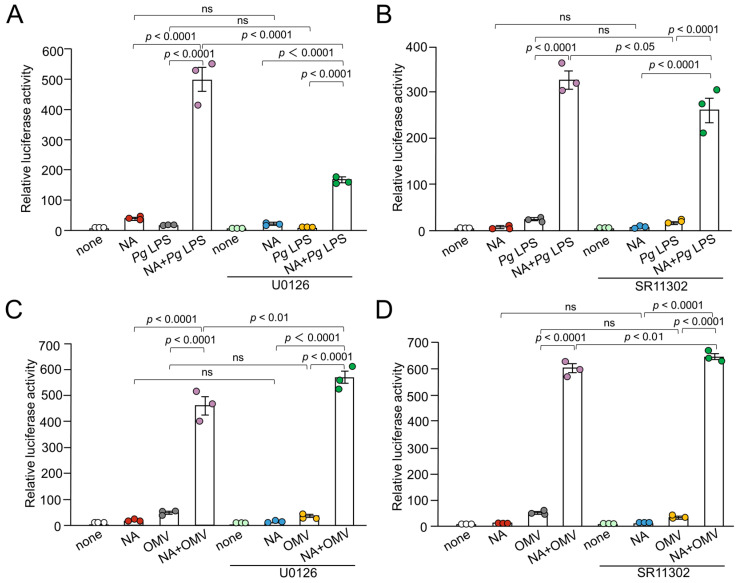
Effects of AP-1 signaling pathway inhibitors on the production of IL-1*β* by BV-2 microglia following treatment with NA, *Pg* virulence factors, and their combination. (**A**,**B**) The mean relative luciferase activity of the IL-1*β* probe induced by NA (1 μM), *Pg* LPS (10 μg/mL), and their combined application in the absence or presence of U0126 (15 μM) (**A**) or SR11302 (10 μM) (**B**). (**C**,**D**) The mean relative luciferase activity of the IL-1*β* probe induced by NA (1 μM), OMVs (150 μg of protein/mL), and their combined application in the absence or presence of U0126 (15 μM) (**C**) or SR11302 (10 μM) (**D**). Data relative to the values are presented as the mean ± SE of 3 independent experiments.

**Figure 6 ijms-26-02660-f006:**
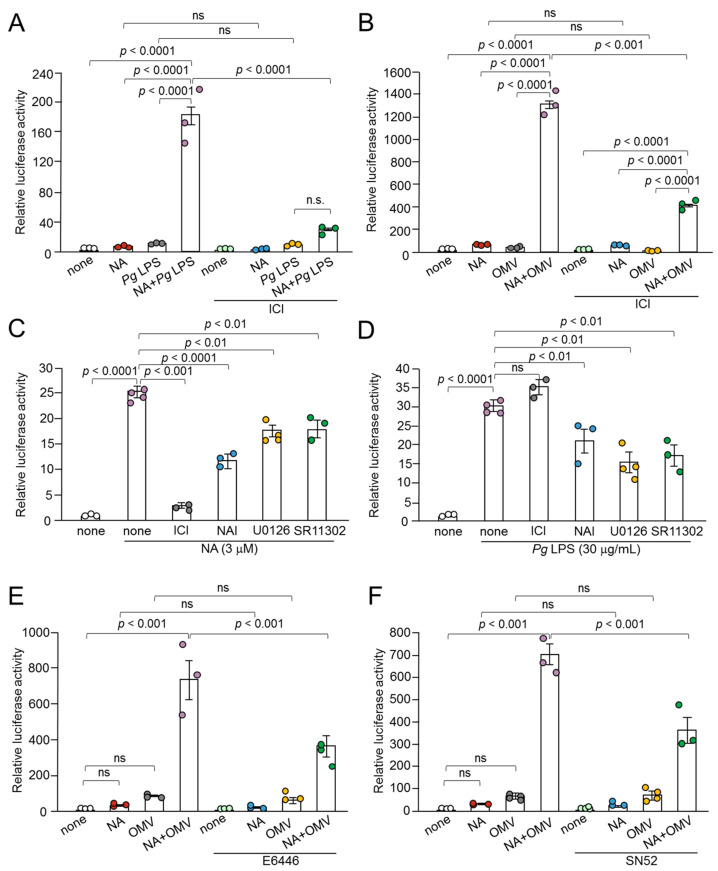
Involvement of A*β*_2_R in the synergistic effect of NA and *Pg* virulence factors in the production of IL-1*β *by BV-2 microglia and effects of signaling inhibitors on the production of IL-1*β *induced by NA or *Pg* LPS alone. (**A**,**B**) Effects of ICI on the production of IL-1*β* by BV-2 microglia following stimulation of NA (1 μM) plus *Pg* LPS (10 μg/mL) or NA (1 μM) plus OMVs (150 μg/mL). (**C**,**D**) Effects of *β*_2_ ICI-118551 (0.1 μM), NAI (100 nM), U0126 (15 μM), and SR11302 (10 μM) on the production of IL-1*β *by BV-2 microglia following treatment with NA (3 μM) or *Pg* LPS (30 μg/mL). (**E**,**F**) Possible involvement of phagocytosis-related pathways on the production of IL-1*β *by BV-2 microglia following treatment with NA (1 μM) plus OMVs (150 μg/mL) was examined using the TLR9 inhibitor E6446 (10 μM) (**E**) or the p52 inhibitor SN52 (40 μg/mL) (**F**). Data relative to the values are presented as the mean ± SE of 3–4 independent experiments.

**Figure 7 ijms-26-02660-f007:**
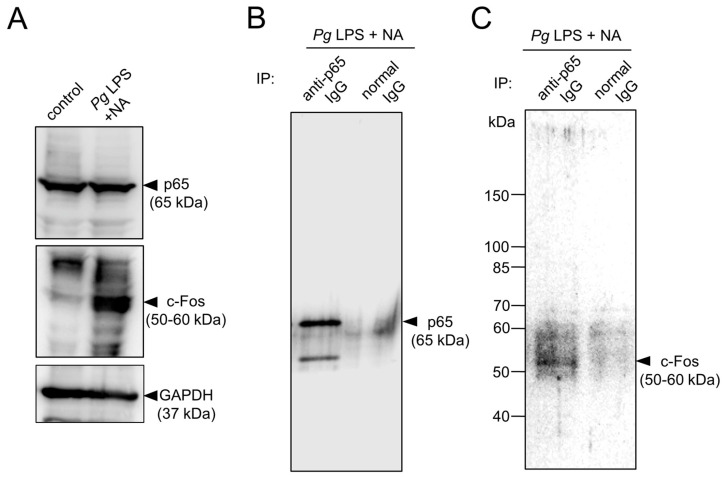
Possible changes in expression levels and protein complex formation of p65 and c-Fos in BV-2 microglia following combined treatment with NA and *Pg* LPS. (**A**) Western blotting to analyze the protein levels of p65 and c-Fos in BV-2 microglia following treatment with NA and *Pg* LPS. Immunoblotting of GAPDH was used as a loading control. (**B**,**C**) Co-immunoprecipitation combined with Western blotting to analyze the interaction between p65 and c-Fos in BV-2 microglia following treatment with NA and *Pg* LPS. Co-immunoprecipitated samples using normal or anti-p65 IgG were subsequently subjected to Western blotting using anti-p65 IgG (**B**) or anti-c-Fos IgG (**C**). Each experiment was repeated at least three times.

**Figure 8 ijms-26-02660-f008:**
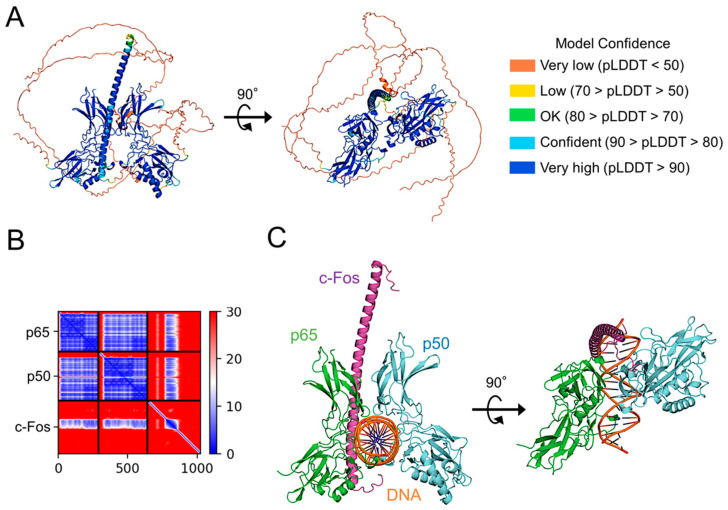
Prediction of c-Fos binding to NF-*κ*B p65:p50 heterodimer. (**A**) The structural model of c-Fos-bound NF-*κ*B p65:p50 heterodimer generated using AlphaFold2, presented as a ribbon colored pLDDT score. c-Fos binds to the molecular surface of p65. The figure to the right shows the complex structure vertically rotated by 90° from the figure to the left. (**B**) PAE plots of the c-Fos and p65:p50 complex model. (**C**) The structural model of the p65:p50:c-Fos complex forms a composite surface for DNA recognition. For the model construction, DNA bound to the p65:50 heterodimer (PDB code 2I9T) was bound to the AlphaFold2 model of the p65:p50:c-Fos complex. The long disordered regions in the *N*-terminus and *C*-terminus of c-Fos were removed from the figure to facilitate visualization. The NF-*κ*B of p65 and p50 are in green and light blue, respectively. c-Fos is in purple. The two DNA strands are in orange. The figure to the right shows the complex structure vertically rotated by 90° from the figure to the left. The figures were drawn using PyMOL 3.1 [27].

## Data Availability

Data are contained within the article.

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
