# Peer review of "Noradrenaline Synergistically Enhances Porphyromonas gingivalis LPS and OMV-Induced Interleukin-1β Production in BV-2 Microglia Through Differential Mechanisms"

_ijms, 2025, doi:10.3390/ijms26062660_

Round 1

Reviewer 1 Report (New Reviewer)

Comments and Suggestions for Authors

Dear Sirs,

The manuscript describes how Pg is involved in local systemic immune and inflammatory responses during progression of Alzheimer’s disease. The authors studied the regulatory mechanisms of noradrenaline on the production of interleukin-1β (IL-1β) by microglia following stimulation with Pg virulence factors, lipopolysaccharide (LPS) and outer membrane vesicles (OMVs). The topic is very interersting and the script well-written. Please correct according my minor comments

Comment 1, lines 34-35. Please add the used doses (LP, OMV and NA) in the abstract.

comment 2, lines 77-88. OMVs should also be introduced ; please balance the intro and discussion parts. What they are as well as their earlier data on OMVs and its effect on inflammation ( plus the abbreviation of OMVs in the text should be open at the first time)

comment 3, lines 97-108. Why the differnet NA levels were used and only one OMV and LPS levels, please explain

comment 4 line 455. Describe shortly how OMV have been isolated and their contration was measured

comment 5, line 416. The number of experiments whould be added as a limited factor,

comment 6 lines 435-433. how many experiments were done should be added in line 435

Comment 7, references. They are quite old. please check the reference list. OMV study is rather novel.

Author Response

Comments 1. lines 34-35. Please add the used doses (LP, OMV and NA) in the abstract.

Response 1: We have added the concentrations of Pg LPS, OMVs and NA in the Abstract section.

Pg LPS: (10 μg/mL), OMVs (150 μg of protein/mL), NA (30-1000 nM)

Comments 2: lines 77-88. OMVs should also be introduced ; please balance the intro and discussion parts. What they are as well as their earlier data on OMVs and its effect on inflammation (plus the abbreviation of OMVs in the text should be open at the first time).

Response 2: Thank you for that pertinent comments. We have added the following sentences in the Introduction section.

Lines 77-80: Besides LPS, Pg produces outer membrane vesicles (OMVs) that contain LPS, proteases including gingipains and pathogen-derived DNA and RNA [1]. OMVs secreted from Pg induced AD-like pathologies in middle-aged mice [10].

1. Liu, S.; Catherine, A.; Butler, C.A.; Ayton, S.; Reynolds, E.C.; Dashper S.G. Porphyromonas gingivalis and the pathogenesis of Alzheimer's disease. Crit. Rev. Microbiol. 2024, 50, 127-137. doi: 10.1080/1040841X.2022.2163613.

10. Gong, T.; Chen, Q.; Mao, H.; Zhang, Y.; Ren, H.; Xu, M.; Chen, H.; Yang, D. Outer membrane vesicles of Porphyromonas gingivalis trigger NLRP3 inflammasome and induce neuroinflammation, tau phosphorylation, and memory dysfunction in mice. Front. Cell. Infect. Microbiol. 2022, 12, 925435. doi: 10.3389/ fcimb.2022.925435.

Comments 3: lines 97-108. Why the different NA levels were used and only one OMV and LPS levels, please explain.

Response 3: We have established that 10 μg/mL and 150 μg of protein/mL are the submaximal concentrations of Pg LPS and OMVs respectively, that can stably induce the significant increase in the luciferase activities of Nanoluc reporter BV-2 microglia (Inoue et al., Cells, 2023). Therefore, we have used Pg LPS (10 μg/mL) and OMVs (150 μg of protein/mL) in this study. On the other hand, we have examined a wide range of NA concentrations (30 – 1000 nM) in this study, because no information is available about the concentrations of NA that can induce the significant increase in the luciferase activities when applied alone or in combination with Pg virulence factors. 

Comments 4: line 455. Describe shortly how OMV have been isolated and their concentration was measured.

Response 4: We have added following sentences in the Materials and Methods section.

Lines 469-475: Briefly, culture mediums of Pg were centrifuged to remove bacterial cells. The supernatant was filtrated with Millipore filter (poresize, 0.22 μm, Millipore, MA, USA), and then ultracentrifuged. The pellet was collected as OMVs and dissolved in PBS. The protein concentration of OMVs were measured by Pierce™ BCA protein assay kit (ThermoFisher Scientific Inc., Waltham, MA, USA) and the concentrations of OMVs were adjusted to 150 μg of protein/mL. 

Comments 5 and 6: Comments 5: line 416. The number of experiments would be added as a limited factor. Comments 6: lines 435-433. how many experiments were done should be added in line 435.

Responses 5 and 6: We do not understand the meaning of the reviewer’s comments 5 and 6, because line 416 is a sentence in the Discussion section, and line 435 is a title in the Materials and Methods section.

line 416:  PG-LPS, cat: tlrl-pglps, InvitroGen), which were obtained by classical methods using

line  435:  4.2. Cell Culture

Therefore, we have just listed the number of each experiment conducted in this study below.

Figure 1A-F: Data relative to the value are presented as the mean±SE of 4 independent experiments. 

Figure 2A-D: Data relative to the value are presented as the mean±SE of 3 independent experiments. 

Figure 3A, B: Data relative to the value are presented as the mean±SE of 4 independent experiments. 

Figure 4A-D: Data relative to the value are presented as the mean±SE of 3-5 independent experiments. 

Figure 5A-D: Data relative to the value are presented as the mean±SE of 3 independent experiments. 

Figure 6A-F: Data relative to the value are presented as the mean±SE of 3-4 independent experiments. 

Figure 7A-C: Each experiment was repeated at least three times.

As described in the Materials and Methods (4.10. Statistical Analyses), the results were analyzed by a one-way analysis of variance (ANOVA) with a post-hoc Tukey’s test using the GraphPad Prism8 (GraphPad Software, Inc., CA, USA) software package. P values of <0.05 were considered to indicate statistical significance

Comments 7: references. They are quite old. please check the reference list. OMV study is rather novel. 

Response 7: We have tried to replace old references with new ones. We would like to cite some old references ([21] Yang et al., 1999; [43] Blasi et al., 1990), because they can not be replaced with new ones.

Ogawa et al., Microbiology 1994 → Ogawa et al., Front. Biosci. 2007

Lamont et al., Microbiol. Mol. Biol. Rev. 1998 → Coats et al., Infect. Immun. 2003

Stein et al., EMBO J. 1993 → Oeckinghaus et al. Nat. Imunol. 2011

Pizon et al., J. Cell Sci. 1994 → Pareja et al. PLos One 2019

4. Ogawa, T.; Asai, Y.; Makimura, Y.; Tamai, R. Chemical structure and immunobiological activity of Porphyromonas gingivalis lipid A. Front. Biosci. 2007, 12, 3795-812. doi: 10.2741/2353. 

5. Coats, S.R.; Reife, R.A.; Bainbridge, B.W.; Pham, T.T.; Darveau, R.P. Porphyromonas gingivalis lipopolysaccharide antagonizes Escherichia coli lipopolysaccharide at toll-like receptor 4 in human endothelial cells. Infect. Immun. 2003, 71, 6799–6807. doi: 10.1128/IAI.71.12.6799-6807.2003 

25. Oeckinghaus, A.; Hayden, M.S.; Ghosh, S. Crosstalk in NF-κB signaling pathways. Nat. Immunol. 2011, 12, 695-708. doi: 10.1038/ni.2065.

41. Pareja, M.E.M.; Gaurón, M.C.; Robledo, E.; Aguilera, O.; Colombo, M.I. The cAMP effectors, Rap2b and EPAC, are involved in the regulation of the development of the Coxiella burnetii containing vacuole by altering the fusogenic capacity of the vacuole. PLos One 2019, 14, e0212202. doi: 10.1371/journal.pone.0212202.

Reviewer 2 Report (New Reviewer)

Comments and Suggestions for Authors

Thank you for the invitation to review this manuscript. It is a well-conducted study that explores the synergistic effects of noradrenaline (NA) on Porphyromonas gingivalis (Pg) virulence factor-induced IL-1β production in microglia, shedding light on differential mechanisms involving NF-κB and AP-1 pathways. Below, I provide a section-by-section:

Abstract and introduction:

- The abstract and introduction could briefly highlight the novelty of the study. For example, it could highlight that this is the first study to show that NA synergistically enhances IL-1β production in response to Pg LPS and OMVs through distinct mechanisms.

- Highlighting the gap in the literature: The introduction could more clearly articulate the specific gaps in understanding that this study aims to address, particularly in relation to the differential mechanisms of NA and Pg virulence factors.

Results and discussion:

- The study mentions the involvement of NF-κB and AP-1, but the interplay between these transcription factors and how they are differentially regulated by NA and Pg virulence factors could be explored further.

- The study should include more information on the reproducibility of the results, such as the number of independent experiments performed and any variability observed.

- Addressing limitations: The discussion should more explicitly address the limitations of the study, such as the use of a single microglial cell line (BV-2) and the potential differences in responses in vivo. The study uses commercially available Porphyromonas gingivalis (Pg) LPS, extracted from the ATCC 33277 strain, which may not fully represent the natural LPS structure due to potential modifications during extraction. These limitations should be acknowledged, and future studies using LPS from more virulent strains or in the context of OMVs are recommended to validate the findings and ensure biological relevance.

- The discussion could benefit from a more detailed exploration of the potential clinical implications of the findings.

- The discussion should suggest specific future research directions.

Materials and Methods:

- The section should provide more details on the culture conditions for BV-2 cells, including the passage number.

- The methods for OMV preparation could be more detailed, including any quality control steps to ensure the purity and consistency of the OMVs. For example, it should be clarified whether the OMVs were characterized for size, protein content, and the presence of specific virulence factors.

Author Response

Comments 1: The abstract and introduction could briefly highlight the novelty of the study. For example, it could highlight that this is the first study to show that NA synergistically enhances IL-1β production in response to Pg LPS and OMVs through distinct mechanisms. 

Response 1: Thank you for the useful suggestion. We have added following sentences in the Abstract and Introduction sections.

the Abstract section:

Lines 48-50: Our study provides the first evidence that NA synergistically enhances the production of IL-1β in response to Pg LPS and OMVs through distinct mechanisms. 

the Introduction section:

Lines 97-98: The present findings highlight that NA synergistically enhances the production of IL-1β in response to Pg LPS and OMVs through distinct mechanisms. 

Comments 2: Highlighting the gap in the literature: The introduction could more clearly articulate the specific gaps in understanding that this study aims to address, particularly in relation to the differential mechanisms of NA and Pg virulence factors. 

Response 2: Thank you pointing them to us. We have added following sentences in the 

Introduction sections.

Lines 97-101: The present findings highlight that NA synergistically enhances the production of IL-1β in response to Pg LPS and OMVs through distinct mechanisms. This study is the first to show the existence of an intrinsic augmentation mechanism of Pg virulence factor-induced neuroinflammation by microglia, providing biological evidence for the impact of stress on the progression of periodontitis-related systemic inflammatory diseases.

Comments 3: The study mentions the involvement of NF-κB and AP-1, but the interplay between these transcription factors and how they are differentially regulated by NA and Pg virulence factors could be explored further. 

Response 3: Thank you for the pertinent comments. We have added following sentences in the Discussion sections.

Lines 419-423: Additional experiments are necessary to further elucidate physical and functional interactions between NF-κB and AP-1 c-Fos, that lead to enhanced transcriptional activity at mainly κB enhancer-dependent promoter. Moreover, the mechanisms that these two transcriptional factors are differentially regulated by NA and Pg virulence factors are to be addressed in future studies.

Comments 4: The study should include more information on the reproducibility of the results, such as the number of independent experiments performed and any variability observed. 

Response 4: We have added the number of experiments in each Figure legend.

Figure 1A-F: Data relative to the value are presented as the mean±SE of 4 independent experiments. 

Figure 2A-D: Data relative to the value are presented as the mean±SE of 3 independent experiments.

Figure 3A, B: Data relative to the value are presented as the mean±SE of 4 independent experiments. 

Figure 4A-D: Data relative to the value are presented as the mean±SE of 3-5 independent experiments. 

Figure 5A-D: Data relative to the value are presented as the mean±SE of 3 independent experiments. 

Figure 6A-F: Data relative to the value are presented as the mean±SE of 3-4 independent experiments. 

Figure 7: Each experiment was repeated at least three times.

As described in the Materials and Methods (4.10. Statistical Analyses), the results were analyzed by a one-way analysis of variance (ANOVA) with a post-hoc Tukey’s test using the GraphPad Prism8 (GraphPad Software, Inc., CA, USA) software package. P values of <0.05 were considered to indicate statistical significance

Comments 5: Addressing limitations: The discussion should more explicitly address the limitations of the study, such as the use of a single microglial cell line (BV-2) and the potential differences in responses in vivo. The study uses commercially available Porphyromonas gingivalis (Pg) LPS, extracted from the ATCC 33277 strain, which may not fully represent the natural LPS structure due to potential modifications during extraction. These limitations should be acknowledged, and future studies using LPS from more virulent strains or in the context of OMVs are recommended to validate the findings and ensure biological relevance. 

Response 5: We appreciate these useful comments. We have reconstructed sentences in the Discussion section in order to expand the description concerning with the limitations of this study as following.

Lines 424-437: The present study is beset with certain limitations. First, all experimental findings were obtained from in vitro studies by use of BV-2 microglia. Therefore, additional experiments using animals subjected to stress are needed to verify the present findings. Second, we used only commercially available Pg LPS (STD PG-LPS, cat: tlrl-pglps, InvitroGen), which were obtained by classical methods using hot-phenol extraction using the bacterial strain ATCC 33277, suggesting that the TLR2-dependent IL-1β production by BV-2 microglia was mediated mainly by concomitant lipoproteins. Furthermore, Pg LPS-lipid A may be received modification of acetylchains and phosphates of lipid A during preparation of Pg LPS, resulting in reduction of TLR4 activating potency. It has been reported that different bacterial strains contain different factors in their own OMVs, which may determine their pathogenicities [42]. ATCC 33277 used in this study is classified as a less-virulent strain. Further studies are also necessary to examine the effects of Pg LPS, lipoproteins and OMVs prepared from several more virulent bacterial strains, including W83 and possibly more virulent clinical isolates, on TLRs of microglia. 

Comments 6: The discussion could benefit from a more detailed exploration of the potential clinical implications of the findings. 

Response 6: We have added more detailed exploration of the potential clinical implications of the findings in the Discussion sections as following. 

Lines 344-347: To the best of our knowledge, this is the first report showing that NA synergizes with Pg virulence factors to up-regulate the expression of IL-1β in the microglia. The present findings may provide biological evidence for the impact of stress on the progression of periodontitis-related systemic inflammatory diseases.

Comments 7: The discussion should suggest specific future research directions. 

Response 7: We have added specific future research directions in the Discussion sections as following. 

Lines 419-423: Additional experiments using animals subjected to stress are necessary to verify the present findings using BV-2 microglia.

Comments 8: The section should provide more details on the culture conditions for BV-2 cells, including the passage number. 

Response 8: We have added a following sentence in the Materials and Methods section.

Lines 452-453: BV-2 cells at passage number 3-10 were used in the experiments. 

Comments 9: The methods for OMV preparation could be more detailed, including any quality control steps to ensure the purity and consistency of the OMVs. For example, it should be clarified whether the OMVs were characterized for size, protein content, and the presence of specific virulence factors. 

Response 9: The diameter of collected OMVs were expected to be under 0.22 μm because we filtrated Pg culture supernatant with 0.22 μm filter before collecting OMVs from them. We measured the protein concentration of the collected OMV solution by BCA protein assay and OMVs were added to BV-2 cells at 150 μg of protein/mL. However, we didn’t check whether collected OMVs contained the specific virulence factors.  

Reviewer 3 Report (New Reviewer)

Comments and Suggestions for Authors

comments in word 

Comments on the Quality of English Language

adequate English quality

Author Response

Comments 1: Introduction Section. Please add more information about Porphyromonas gingivalis, including its general characteristics, the diseases it causes, and its impact on oral health. 

Response 1: Thank you for the suggestions.We have reconstructed sentences concerning with information about Porphyromonas gingivalis in the Introduction section as following.

Lines 56-59: Porphylomonas gingivalis (Pg) is a Gram-negative bacterium as a major pathogen of periodontal disease [1]. Accumulating epidemiological evidence indicates that infection with  Pg causes a large number of systemic diseases, including diabetes mellitus, rheumatoid arthritis and Alzheimer’s disease (AD).

Comments 2: Materials and Methods section. Ensure that all reagents and equipment include the brand and country of origin. 

Response 2: We have checked the brand and country of all reagents and equipments used in this study.

Line 439: 5% fetal bovine serum (Nichirei, Tokyo, Japan)

Line 439: penicillin and streptomycin (Fujifilm, Tokyo, Japan)

Line 445: 96-well white culture plate (IWAKI, Tokyo, Japan) 

Line 473: 6 cm dish (Sumitomo Bakelite, Tokyo, Japan)

Line 482: 10 cm dish (IWAKI, Tokyo, Japan)

Line 489: PDVF membrane (Merck, MA, USA)

Comments 3: Results Section. All figures are clear and well-presented. 

Response 3: Thank you.

Comments 4: References section. Verify that all references follow the correct format required by the journal. 

Response 4: We have verified that all references follow the correct format required by the journal. 

Comments 5: Discussion Section. Add a brief paragraph highlighting the importance and relevance of this study. 

Response 5: Thank you for the useful comment. We have reconstructed sentences highlighting the importance and relevance of this study in the Discussion section as following. 

Lines 344-347: To the best of our knowledge, this is the first report showing that NA synergizes with Pg virulence factors to up-regulate the expression of IL-1β in the microglia. The present findings may provide biological evidence for the impact of stress on the progression of periodontitis-related systemic inflammatory diseases.

Comments 6: Conclusion Section. Include a short statement about the impact of this research. Specify the areas it benefits. the contributions it makes, and what aspects of this study are original compared to others.

Response 6: Thank you for pointing out them to us. We have reconstructed sentences concerning with the impact of this research in the Conclusion section as following. 

Lines 535-539: The present study is the first to show that stress-induced upregulation of NA and Pg vilulence factors can synergistically augment the production of IL-1β by microglia. This may explain the reason why stress influences the progression of periodontitis-related systemic inflammatory diseases.

Round 2

Reviewer 2 Report (New Reviewer)

Comments and Suggestions for Authors

The authors have adequately addressed the reviewer’s comments. I recommend the acceptance of the manuscript.

This manuscript is a resubmission of an earlier submission. The following is a list of the peer review reports and author responses from that submission.

Round 1

Reviewer 1 Report

Comments and Suggestions for Authors

The article by Sakura Muramoto and colleagues details how Noradrenaline Synergistically Enhances Porphyromonas gingivalis LPS and OMv-induced interleukin-1β Production in BV-2 Microglia Through Differential Mechanisms. In this article, the authors have studied the possible functional interaction between NA and Pg virulence factors in producing IL-1β by BV-2 microglia. The overall quality of the article must be improved, and the following changes should be made.

4.7. Immunoblotting BV-2 microglia were seeded in a 6 cm petri dish at a density of 3.3 106 cells/dish for 1 day. Rewrite this sentence for clarity. The authors should make necessary changes throughout the article. 

Lines 89, 114, 116, 336, and 451 need minor corrections. 

The Discussion section lacks specific literature references while discussing the study's results. The discussion section must include the relevant studies.

Figure 3C and 3D should include the molecular weight for each protein represented in the figure. The signal shown in the blots is saturated and needs to be replaced with another blot. The western blot analysis should be done and included in this figure. The GAPDH expression level should be similar.

Figures 7C and 7D should be replaced. The figure does not show clear bands for p65 and c-Fos. The western blot analysis should be done and included in this figure. 

The article needs significant changes in terms of grammatical errors, spelling mistakes, and scientific nomenclature.

The conclusion section can be added.

Author Response

1.  Lines 436-437:  BV-2 microglia were seeded in a 6 cm petri dish at a density of 

3.3 106 cells/dish for 1 day. Rewrite this sentence for clarity. The authors should make necessary changes throughout the article.

[Author’s Reply]  Corrected through the article.

2. Lines 89, 114, 116, 336, and 451 need minor corrections.

[Author’s Reply]  Corrected.

3. The Discussion section lacks specific literature references while discussing the study's results. The discussion section must include the relevant studies.

[Author’s Reply]

According to the reviewer’s comment, we have added 3 specific literature references in the Discussion section. 

  1. Asai, Y.; Hashimoto, M.; Fletcher, H.M.; Miyake, K.; Akira, S.; Ogawa, T. Lipopolysaccharide preparation extracted from Porphyromonas gingivalis lipoprotein-deficient mutant shows marked decrease in toll-like receptor 2-mediated signaling. Infect. Immun. 2005, 73, 2157-2163.
  2. Nativel, B.; Couret, D.; Giraud, P.; Meïlhac, O.; d’Hellencout, C.L.; Viranaïcken, W.; Da Silva, C.R. Porphyromonas gingivalis lipopolysaccharides act eclusively through TLR4 with a resilience between mouse and human. Sci. Rep. 2017, 7:15789. 
  1. Mantri, C.K.; Chen, C.H.; Dong, X.; Goodwin, J.S.; Pratap, S.; Paromov, V.; Xie, H. Fimbriae-mediated outer membrane vesicle production and invasion of Porphyromonas gingivalis. Microbiologyopen 2015, 4, 53-65. 

4. Figure 3C and 3D should include the molecular weight for each protein represented in the figure. The signal shown in the blots is saturated and needs to be replaced with another blot. The western blot analysis should be done and included in this figure. The GAPDH expression level should be similar.

[Author’s Reply]

According to the reviewer’s comment, we have replaced a western blot in the top panel of Fig.3C with another blot obtained with shorter exposure time for chemiluminescence detection with an X-ray imager. We have also added the molecular weight for each protein represented in Figure 3C and 3D. 

5. Figures 7C and 7D should be replaced. The figure does not show clear bands for p65 and c-Fos. The western blot analysis should be done and included in this figure.

[Author’s Reply]

In Figure 7C, a band corresponding to p65 was clearly visible in the lane for anti-p65 IgG, but not in the lane for normal IgG. In Figure 7D, however, rather multiple smeared bands at 50-60 kDa were visible in the lane for anti-p65 antibody, but not in the lane for normal IgG, probably because a rabbit polyclonal antibody for c-Fos (ab190289, Abcam), which detects multiple bands at 50-60 kDa corresponding to different isoforms of c-Fos, was used in this study. It is also considered that dissociation of protein complex consisting of c-Fos and p65 during the experimental procedure may be responsible for the smeared band of c-Fos in the co-immunoprecipitated samples.

We have added necessary descriptions in the Result section.

6. The article needs significant changes in terms of grammatical errors, spelling mistakes, and scientific nomenclature.

[Author’s Reply]  Corrected.

7. The conclusion section can be added.

[Author’s Reply]  We have added the conclusion section.

Reviewer 2 Report

Comments and Suggestions for Authors

In their original article entitled „Noradrenaline (NA) Synergistically Enhances Porphyromonas gingivalis LPS and OMV-Induced Interleukin-1β Production in BV-2 Microglia Through Differential Mechanisms” Muramoto et al. present evidences that NA may co-stimulate Il-1ß production in microglia cells after co-stimulation with "standard" Pg-LPS and Pg-OMV.

General comments: There is hardly any true standard of Pg-LPS and Pg-OMV, even if the preparations are form the ATCC-type strain. The Pg diversity of virulence and factors associated (including LPS and OMV) is just too large. This is a strong limitation which needs to be discussed.

Please delete the many extra blanks in text.

Abstract

“Pg and its virulence factor lipopolysaccharide (LPS) have low biological activities.” I and many other oral microbiologist do strongly disagree. This sentence is a kind of provocative and must substantially be defended or –better- deleted, Pg is the key-pathobiont in periodontal diseases and HAS high biological activities (see below, e.g. it's TLR-4 antagonistic role). 

Introduction

Line 53ff: Please check that Porphyromonas and P. gingivalis (and Pg if you prefer) is written in italics throughout the text and legends.

“development” is doubled in line 61.

Line 66: “Furthermore, Pg LPS is a weaker Toll-like receptor (TLR) 2/4 agonist” yes, might be true for the commercial Pg LPS standard, but usually Pg LPS from more virulent strains is instead a TLR-4 antagonist, blocking killing. This is due to de-acetylation and de-phosphorylation of it's Lipid A (what you explain next sentence). This competitive inhibition is a major biological role of the clinically important Pg LPS!

Line 68: you mean “than is E. coli” I guess?

Lines 71-72: “However, the mechanisms by which Pg LPS can augment local systemic immune and inflammatory re-sponses remains unclear.” Check literature such as 10.1111/j.1462-5822.2007.00935.x where it is written “This [TLR4]-antagonistic activity of H. pylori or P. gingivalis LPS, as well as their TLR2 activation capability may be associated with their ability to contribute to atherosclerosis [and other systemic diseases]." Of course, a lot more is known about the mechanisms already.

Results

Line 151 : separate “ofboth”

The Results-section is otherwise well written with professional graphs and statistics. By contents it is plausible but this reviewer is not specialized enough to review the immunological-neuropathological part in depths.

Discussion

Line 328: please correct “E .coli” to “E. coli”

Line 336: “lucife rase activity”; please correct

The Discussion-section is otherwise well written with representative literature cited. What is missing is a paragraph about the limitations due to application of only one ("Standard") of (weak!) Pg LPS and OMV. See below. 

Methods

OMV: "Pg ATCC33277 was maintained as previously described [24]. OMVs were prepared by ultracentrifugation as previously described": In Pg we see quite heterogeneity on all levels of secretome and virulence factors. The ATCC is still a reference strain but does not cover the heterogeneity of OMV- and their possible content and capacity at all. See literature (f.i. about OMV content of ATCC versus W83 in Mantri et al. 2015, Microbiology open).

More important and a major issue in the context: Standard Pg LPS, a TLR4/TLR2 agonist, was purchased from InvivoGen (San Diego, CA, USA). This is indeed what the manufacturer tell about their LPS. As a matter of fact, Pg can produce different variants of LPS (depending on no. of acetylchains and phosphates of Lipid A). A “standard” of Pg LPS does not exit, especially not really representative for the diversity of strains. The best way would be to prepare own Pg LPS fractions from several more virulent clinical isolates but this is a very difficult method, I know. So please discuss this limit in more depths. And again; the biological role of this weak (!) standard Pg LPS might be small but there are very potent Pg LPS derivates with TLR4 antagonistic effects including support of the entire Gram-negative red/orange complex of pathobionts as their TLR4-derived killing is blocked.

Author Response

1. Abstract

(1) “Pg and its virulence factor lipopolysaccharide (LPS) have low biological activities.” I and many other oral microbiologist do strongly disagree. This sentence is a kind of provocative and must substantially be defended or –better- deleted, Pg is the key-pathobiont in periodontal diseases and HAS high biological activities (see below, e.g. it's TLR-4 antagonistic role). 

[Author’s Responses]

We appreciate the reviewer for the pertinent comment. According to the reviewer’s comment, we have deleted “Pg and its virulence factor lipopolysaccharide (LPS) have low biological activities.” We have reconstructed sentences as following:

“However, it is not yet fully understood how Pg can augment local systemic immune and inflammatory responses especially during progression of AD.”

(2) Line 53: Please check that Porphyromonas and P. gingivalis (and Pg if you prefer) is written in italics throughout the text and legends.

[Author’s Responses]  Corrected throughout the text and legends. 

(3) “development” is doubled in line 61.

[Author’s Responses]  Corrected.

(4) Line 66: “Furthermore, Pg LPS is a weaker Toll-like receptor (TLR) 2/4 agonist” yes, might be true for the commercial Pg LPS standard, but usually Pg LPS from more virulent strains is instead a TLR-4 antagonist, blocking killing. This is due to de-acetylation and dephosphorylation of it's Lipid A (what you explain next sentence). This competitive inhibition is a major biological role of the clinically important Pg LPS!

[Author’s Responses]

I totally agree with the reviewer that Pg is the key-pathobiont in periodontal diseases and has high biological activity. However, accumulating evidence shows that Pg LPS appears to be a poor activator of monocyte production of proinflammatory cytokines compared with E. coli LPS. According to the reviewer’s comments, we have reconstructed sentences as following:

“Moreover, Pg can influence the development of these periodontitis-related systemic disorders by affecting the adaptive immune response and inducing inflammation and innate immune responses [1]. In contrast, Pg lipopolysaccharide (LPS) is a relatively poor inducer of monocyte production of proinflammatory cytokines compared with E. coli LPS [3, 4]. Pg LPS stimulated an inflammatory response when injected into connective tissue but was minimally stimulatory when a systemic response was measured [5, 6]. Furthermore, Pg LPS is a weaker Toll-like receptor (TLR) 2/4 agonist and NF-κB/STAT signaling activator in BV-2 microglia compared with E. coli LPS [7]. The presence of fewer acyl chains and phosphate groups in Pg LPS-lipid A than E. coli-lipid A may be responsible for a lower level of TLR activating potency of Pg LPS at least in BV-2 microglia. Furthermore, it was also reported that Pg LPS exerted antagonistic effects toward TLR4-dependent cell activation by E. coli LPS [8]. These properties may allow Pg to evade innate host defense mechanisms without endotoxin tolerance, leading to chronic inflammation. However, it is not yet fully understood the mechanisms by which Pg LPS can augment local systemic immune and inflammatory responses during progression of AD.”

(5) Line 68: you mean “than is E. coli” I guess?

[Author’s Responses]  Corrected.

(6) Lines 71-72: “However, the mechanisms by which Pg LPS can augment local systemic immune and inflammatory responses remains unclear.” Check literature such as10.1111/j.1462-5822.2007.00935.x where it is written “This [TLR4]-antagonistic activity of H. pylori or P. gingivalis LPS, as well as their TLR2 activation capability may be associated with their ability to contribute to atherosclerosis [and other systemic diseases]." Of course, a lot more is known about the mechanisms already.

[Author’s Responses]

We thank the reviewer for this comment. However, we would like to focus on the mechanisms by which Pg LPS induces an intense inflammation in the brain. Therefore, we have reconstructed sentences in the Introduction section as following:

“However, it is not yet fully understood the mechanisms by which Pg LPS can augment local systemic immune and inflammatory responses during progression of AD.”

(7) Line 151 : separate “ofboth”

[Author’s Responses]  Corrected.

(8) The Results-section is otherwise well written with professional graphs and statistics. By contents it is plausible but this reviewer is not specialized enough to review the immunological-neuropathological part in depths.

[Author’s Responses]

Thank you for these comments. In this study, we conducted in vitro experiments using BV-2 microglia. Therefore, further studies are necessary to elucidate immunological-neuropathological changes following chronic systemic exposure of Pg LPS and OMVs in mice under non-stressed and stressed conditions. 

3. Discussion

(1) Line 328: please correct “E .coli” to “E. coli”

[Author’s Responses]  Corrected.

(2) Line 336: “lucife rase activity”; please correct

[Author’s Responses]  Corrected.

(3) The Discussion-section is otherwise well written with representative literature cited. What is missing is a paragraph about the limitations due to application of only one ("Standard") of (weak!) Pg LPS and OMV. 

[Author’s Responses]

According to the reviewer’s comments, we have added sentences with regard to the limitation of this study in the Discussion section as following:

“The limitation of this study is that we used only commercially available Pg LPS (STD PG-LPS, cat: tlrl-pglps, InvitroGen), which were obtained by classical methods using hot-phenol extraction using the bacterial strain ATCC 33277. Therefore, it is considered that the TLR2-dependent IL-1β production by BV-2 microglia was mediated mainly by concomitant lipoproteins. It is also considered that Pg LPS-lipid A may be received modification of acetylchains and phosphates of lipid A during preparation of Pg LPS, resulting in reduction of TLR4 activating potency. Moreover, ATCC 33277 is classified as a less-virulent strain. It has been reported that different bacterial strains contain different factors in their own OMVs, which may determine their pathogenicities [39]. Further studies are necessary to examine the effects of Pg LPS, lipoproteins and OMVs prepared from several more virulent bacterial strains, including W83 and possibly more virulent clinical isolates, on TLRs of microglia.”

  1. Mantri, C.K.; Chen, C.H.; Dong, X.; Goodwin, J.S.; Pratap, S.; Paromov, V.; Xie, H. Fimbriae-mediated outer membrane vesicle production and invasion of Porphyromonas gingivalis. Microbiologyopen 2015, 4, 53-65. 

4. Methods

OMV: "Pg ATCC33277 was maintained as previously described [24]. OMVs were prepared by ultracentrifugation as previously described": In Pg we see quite heterogeneity on all levels of secretome and virulence factors. The ATCC is still a reference strain but does not cover the heterogeneity of OMV- and their possible content and capacity at all. See literature (f.i. about OMV content of ATCC versus W83 in Mantri et al. 2015, Microbiology open). More important and a major issue in the context: Standard Pg LPS, a TLR4/TLR2 agonist, was purchased from InvivoGen (San Diego, CA, USA). This is indeed what the manufacturer tell about their LPS. As a matter of fact, Pg can produce different variants of LPS (depending on no. of acetylchains and phosphates of Lipid A). A “standard” of Pg LPS does not exit, especially not really representative for the diversity of strains. The best way would be to prepare own Pg LPS fractions from several more virulent clinical isolates but this is a very difficult method, I know. So please discuss this limit in more depths. And again; the biological role of this weak (!) standard Pg LPS might be small but there are very potent Pg LPS derivates with TLR4 antagonistic effects including support of the entire Gram-negative red/orange complex of pathobionts as their TLR4-derived killing is blocked.

[Author’s Responses]

We appreciate the reviewer for the pertinent comments. According to the reviewer’s comments, we have added necessary descriptions about TLR4 antagonistic effects of Pg LPS in the Introduction section. We have also added sentences with regard to the limitation of this study in the Discussion section as following: 

Introduction:

“Furthermore, it was also reported that Pg LPS exerted antagonistic effects toward TLR4-dependent cell activation by E. coli LPS [8].”

8. Yoshimura, A.; Kaneko, T.; Kato, Y.; Golenbock, D.T.; Hara, Y. Lipopolysaccharides from periodontopathic bacteria Porphyromonas gingivalis and Capnocytophaga ochracea are antagonists for human toll-like receptor 4. Infect. Immun. 2002, 70, 218-225. 

Discussion:

“The limitation of this study is that we used only commercially available Pg LPS (STD PG-LPS, cat: tlrl-pglps, InvitroGen), which were obtained by classical methods using hot-phenol extraction using the bacterial strain ATCC 33277. Therefore, it is considered that the TLR2-dependent IL-1β production by BV-2 microglia was mediated mainly by concomitant lipoproteins. It is also considered that Pg LPS-lipid A may be received modification of acetylchains and phosphates of lipid A during preparation of Pg LPS, resulting in reduction of TLR4 activating potency. Moreover, ATCC 33277 is classified as a less-virulent strain. It has been reported that different bacterial strains contain different factors in their own OMVs, which may determine their pathogenicities [39]. Further studies are necessary to examine the effects of Pg LPS, lipoproteins and OMVs prepared from several more virulent bacterial strains, including W83 and possibly more virulent clinical isolates, on TLRs of microglia.”

  1. Mantri, C.K.; Chen, C.H.; Dong, X.; Goodwin, J.S.; Pratap, S.; Paromov, V.; Xie, H. Fimbriae-mediated outer membrane vesicle production and invasion of Porphyromonas gingivalis. Microbiologyopen 2015, 4, 53-65.